# ROYAL SOCIETY
# OPEN SCIENCE

electrical engineering/power and energy systems

electric power processing, power packet, logic operation, error correction, cyber-physical system

**Author for correspondence:**
Takashi Hikihara
e-mail: hikihara.takashi.2n@kyoto-u.ac.jp

# Electric power processing using logic operation and error correction

## Shota Inagaki, Shiu Mochiyama and Takashi Hikihara

Department of Electrical Engineering, Kyoto University, Katsura, Nishikyo, Kyoto 615-8510, Japan

SI, 0000-0001-9361-1158; SM, 0000-0001-7126-9187;
TH, 0000-0002-0029-4358

In this study, electric power is processed using the logic operation method and the error correction algorithms to meet load demand. Electric power was treated as the physical flow through the distribution network, which was governed by circuit configuration and efficiency. The hardware required to digitize or packetize electric power, which is called power packet router, was developed in this research work for low power distribution. It provides an opportunity for functional electric power dispatching while disregarding the power flow in the circuit. This study proposes a new design for the network, which makes the logic operation of electric power possible and provides an algorithm to correct the inaccuracies caused by dissipation and noise. Phase shift of the power supply network is resulted by implementing the introduced design.

## 1. Introduction

Electric power is considered as a continually flowing quantity caused by potential difference between nodes in an electric circuit. The smart grid concept interconnects power engineering with information and communication technology (ICT). It also acknowledges the regulation of the power flow between power sources and loads. However, power keeps continually flowing according to an electric circuit configuration even when load demand is quickly and forcibly altered by ICT systems. This fact in turn might lead to a spillover of demands. Hence, in cyber-physical systems, simultaneous operation of the electric network and the ICT system is essential.

Conventional AC power transmission system is an interesting complex network [1,2]. The power network maintains the rule of just-in-time for the system stability and regulation purposes based on dynamics and efficiency. The time constant of the system dynamics is usually long enough to satisfy the temporal change of load demands at the edge. The supply and demand discrepancy is absorbed in huge capacity of linked generators to keep the

## PUBLISHING

equilibrium between the generators in the vicinity. Here, we will reconsider the demand and response system at low power distribution, especially in the closed and isolated systems.

This paper focuses the power distribution in a local grid of a closed system isolated from the commercial power grid, e.g. vehicles, robots, aeroplanes and satellites. In such systems, it is difficult to adopt the aforementioned method for huge power systems. Closed systems have several power sources and batteries, including regeneration from loads that must be operated simultaneously. Thus, there is a limitation of power capacity and the request for real-time power control. In these systems, the information and power must be supplied to the edge without conflict.

In the 1990s, two significant studies were reported in the literature: one was a method to trace the electric power flow in the network [3] and the other was the power packet dispatching method [4]. A smart grid concept provides an opportunity to implement the former method [5], and recently the peer-to-peer technology along with the blockchain is introduced to gain the incentive of employing renewable energy sources [6]. These methods intend to alleviate the problem of mismatch between physical power delivery and cyber analysis.

However, the concept of power packet dispatching was too early to be recognized in the 1990s, because energy storage systems, power-switching devices, and ICT were insufficiently mature. Since the late 2000s, the concept has been studied theoretically and experimentally [7–12]. Besides theoretical considerations, the development of bidirectional routers in kW order [13], the congestion management of the power packet dispatching [14], and the kW class motor drives [15] were achieved. These studies revealed that there is inadequate interface to link the cyber systems and operation of an electric power network. From the viewpoint of applications, the power packet dispatching system enables us to reduce the wire connections between the power sources and loads depending on the change of demand and supply. It will be an advantage for the weight reduction of the harness in the closed systems. In addition, the system accepts the multipower sources in the single system to mitigate the increase of demands or to ensure safety without the new layout of the harness.

This study proposes and confirms *power processing* using logic operation and error correction algorithm, which is not the same as the information processing [16]. The reason lies in the fact that power is not a symbol but a real physical quantity. Detailed discussion as to the discrepancy between power processing and information processing has already been reported in [17]. It confirms that the power packet is logically processed by the unary and binary symbols corresponding to the existence of the power payload attached with an information tag. Furthermore, the payload of power must obey the law of conservation of energy and the law of dissipation. Here, it is crucial to maintain simultaneousness of information and power flows.

This study follows the definition of the logic operation of the power packet dispatching and the algorithm of error correction. The proposed method creates a completely different power distribution system, which disregards the power flow governed by the circuit configuration.

## 2. The logic operation of electric power

Figure 1*a* shows the structure of the power packet. It consists of an information tag and a payload of power. The information tag includes the header and the footer. They are attached as a voltage waveform without current. It means that the tag will not request the circuit configuration between source and load. The routers send and receive the information at sending the power packet. They transfer the payload as a pulsed power with load current, which is decided by circuit configuration. Like the concept, the header conveys a destination address for the payload of power, the requested operation to the payload, and the control codes. The footer conveys an end control signal of the packet.

Figure 1*b* shows the schematic view of a router. The router selects one of the inputs when the left switch turns on. After the switch turns on and connects the source, the capacitor stores the power. The power is transmitted to the output by closing the right and opening left switch set in the desired direction. These switches are controlled according to the received information tag of the packet. Therefore, each power packet is initially tagged to the destination and the request of control. Then, the output ports are selected according to the information written on the header. The previous research works were carried out based on the setting. Figure 1*c* shows the schematic of a power packet dispatching system, where the mixer generates a power packet and a router dispatches it according to the header.

The output side of the router exactly coincides with a mixer. The packetized power is dispatched by using the method of time division multiplex (TDM). The modulation method gives room to avoid the

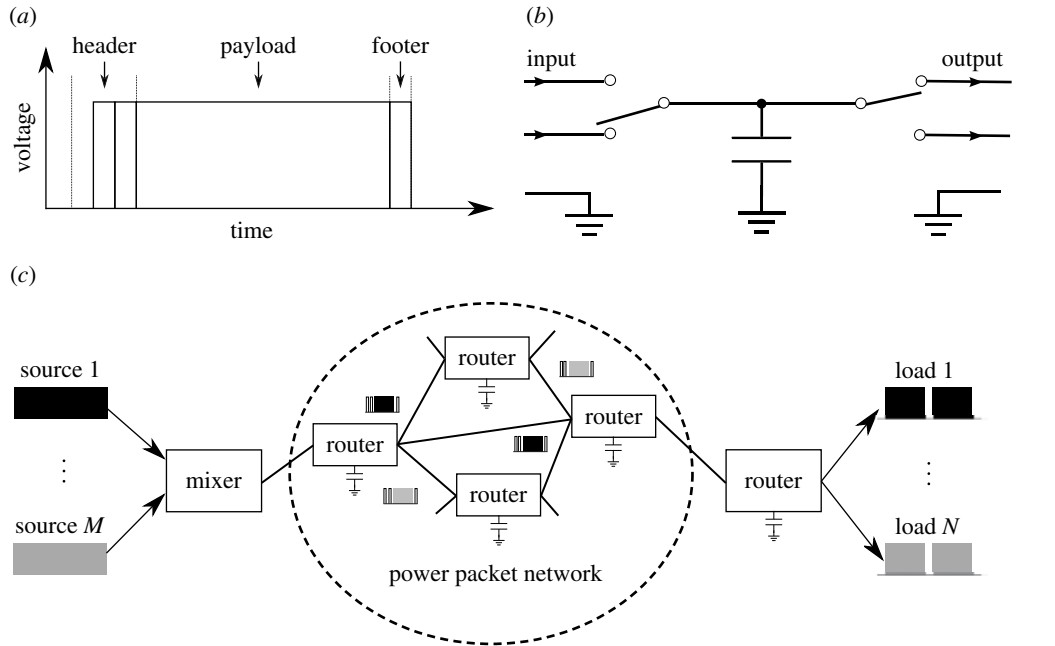

**Figure 1.** Power packet and its dispatching system: (*a*) basic structure of the power packet, (*b*) schematic view of a router and (*c*) power packet dispatching system.

collision of the power packets. The rule of packetization was proposed in [9] based on the optimal dynamic quantizers for discrete-valued input control [17,18]. The rule has already been verified on motor drive in a robot arm with a feedback loop [19,20].

The logic operation is defined at each time slot concerning the existence (1) or non-existence (0) of the power packet according to the voltage amplitude of the payload compared to the threshold voltage. That is, '1' denotes the existence of the payload higher than the threshold voltage and '0' denotes the existence of the payload lower than the threshold. As we set the threshold voltage at $\varepsilon \sim 0$ V higher than the noise, it shows the existence of a power packet.

The unary operation is applied to the input during a time slot $T$ of the payload.

In this study, logic operation of power packets is defined as a function of the router. Logic operations are physically realized as unary (one-bit) or binary (two-bit) operation of power packets. The unary logic operations imply 'NOT' and 'Through'. The binary operations correspond to 'AND', 'OR', 'NAND' and so on. The logic of the signal defines the output of the operations as a logic signal '0' or '1'. However, *power processing* in this research is introduced as the logic operations of power depending on the existence of power. Then, the output becomes logically operated power. Let us assume that every power packet is handled synchronously at the same clock signal. The asynchronous packet transfer has already been discussed in [21]. In this study, command of operation is set on the header of the power packet.

The steps of a logic operation decided by the number of operands are shown here. As for the unary operation, output is decided in the time slot, simultaneously. However, for the binary operation, the output is postponed until a pair of two consecutive time slots are filled. Let us index them to f(orward) and b(ackward). Figure 2*b* describes the binary operation. The output is discharged at the time slot b. This is the fundamental binary operation. There are plenty of variations of the operation depending on the number of ports and the time sequence of TDM.

The definition of the input/output logic relationship follows conventional operations. Figure 2*a* shows the unary non-inversion operation 'Through' and inversion one 'NOT'. Figure 2*b* shows the binary operation 'NAND' and 'AND'. In the logic operation, each voltage is held until the end of an operation. However, the operations' results do not hold the electric power carried by the input power packet because of the circuit configuration.

Electric power is given by temporal integration of instantaneous power by voltage times current. It appears only at the payload in a packet. The logic operation of power requires us to set an additional procedure using an *energy buffer*. An example of 'NAND' operation is given in this study. Table 1 shows the input/output relationship and the usage of the buffer for 'NAND'. The output is given at the backward time slot b. The input '1' at forward time slot f is always stored.

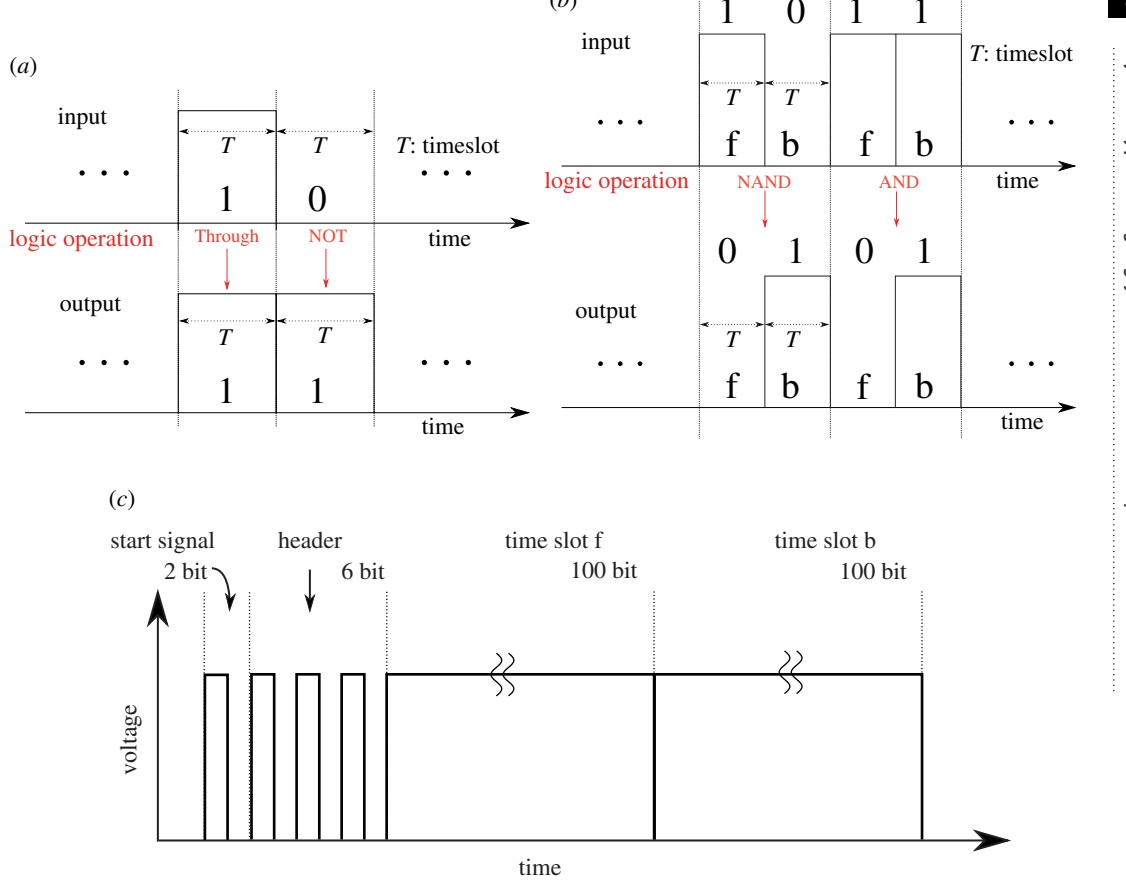

**Figure 2.** Allocation of logic value: (*a*) unary operation, (*b*) binary operations and (*c*) packet for logic operation.

**Table 1.** Truth table of NAND.

| input | | output | | buffer | |
|---|---|---|---|---|---|
| f | b | f | b | f | b |
| 0 | 0 | — | 1 | — | discharge |
| 0 | 1 | — | 1 | — | — |
| 1 | 1 | — | 0 | store | store |
| 1 | 0 | — | 1 | store | discharge |

When the input is '1' at the time slot b and the calculated output becomes '0', the input power is stored. In addition, when the input is '0' at the time slot b, and the output becomes '1', the buffer is discharged. It satisfies that the power consistently flows according to the conventional logic rules. After the logic operation, the structure of the new power packet is configured as in figure 2*c*, and follows TDM manners.

# 3. Verification of static logic operation by router

The logic operation of power is verified by conducting experiments and simulation of the circuit, as shown in figure 3*a*. The mixer generates power packets and dispatches them to the router. The router gets the tag information and executes the logic operation designated by the header.

The load is set at 10 Ω and connected to the router output. The switching elements are composed of Si metal–oxide–semiconductor field-effect transistors (MOSFETs) with 16 mΩ on-resistance, and SiC

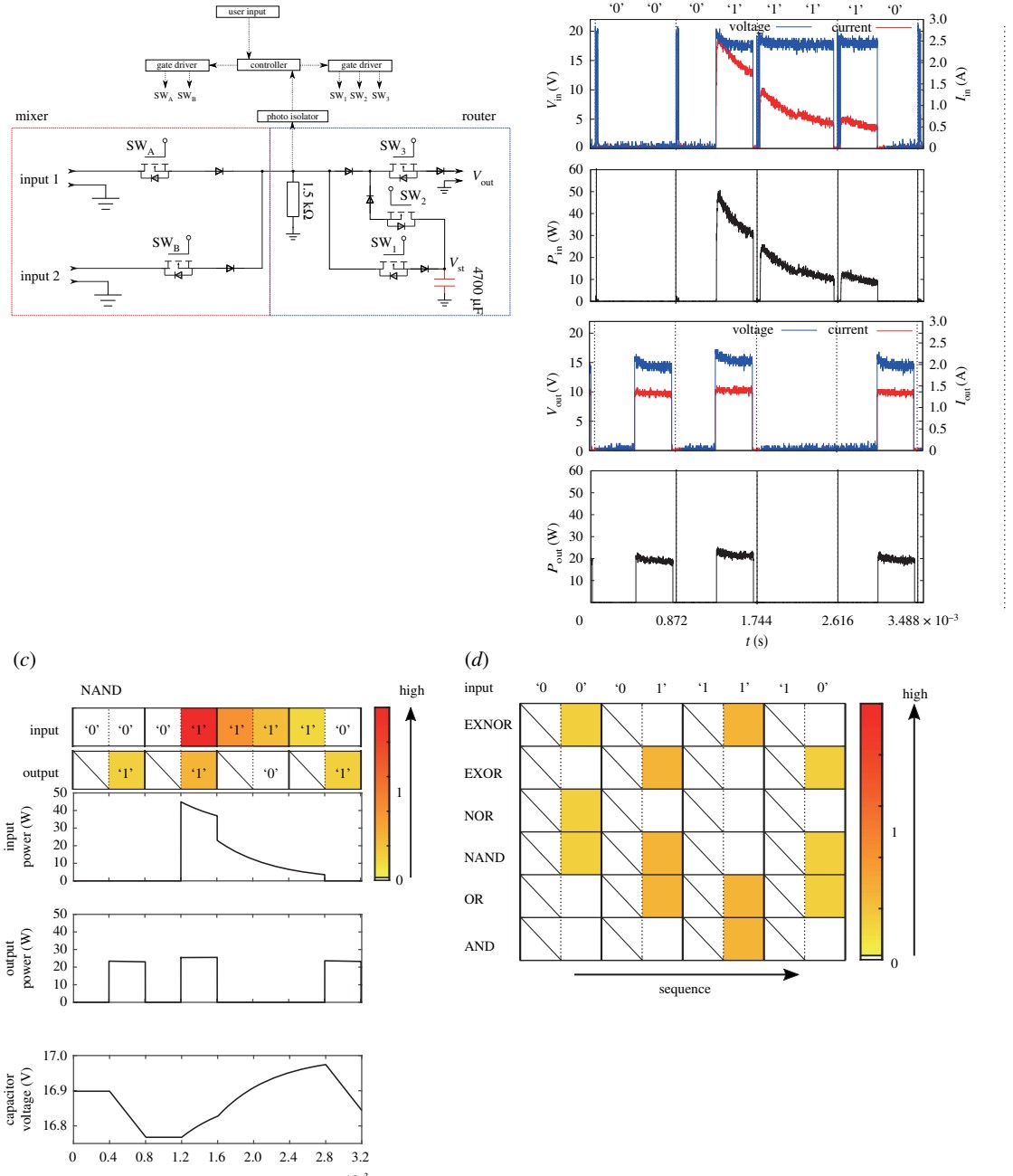

**Figure 3.** Logic operation of the electric power: (*a*) experimental circuit for logic operation, (*b*) NAND operation by experimental circuit in (*a*), and (*c*) simulated switching pattern of NAND operation (input power, output power and voltage of capacitor in the circuit in (*a*)). The colour bar depicts the energy level at each time interval. The energy is normalized by the maximum value of the input energy. (*d*) Output power of possible logical operations of the power packets.

Schottky barrier diodes (SBDs) with 0.8 V forward voltage drop. ZYBO Zynq-7000 development board is adapted as the controller of the switches. The power source is set at 18.4 V DC and the storage is already charged.

Figure 3*b* demonstrates the experimental results of the NAND operation. Depending on the inputs of the power packet, the logic operation is carried out. The experimental results are verified by simulation of the same circuit in figure 3*a*. The current exactly flows according to the power packet. The null payload and the tags do not accept current flow. However, the power is packetized and NAND operation is achieved as the binary operation in figure 3*b*. The logic operation of power packets achieves not only the signal operation but also the logic operation of power.

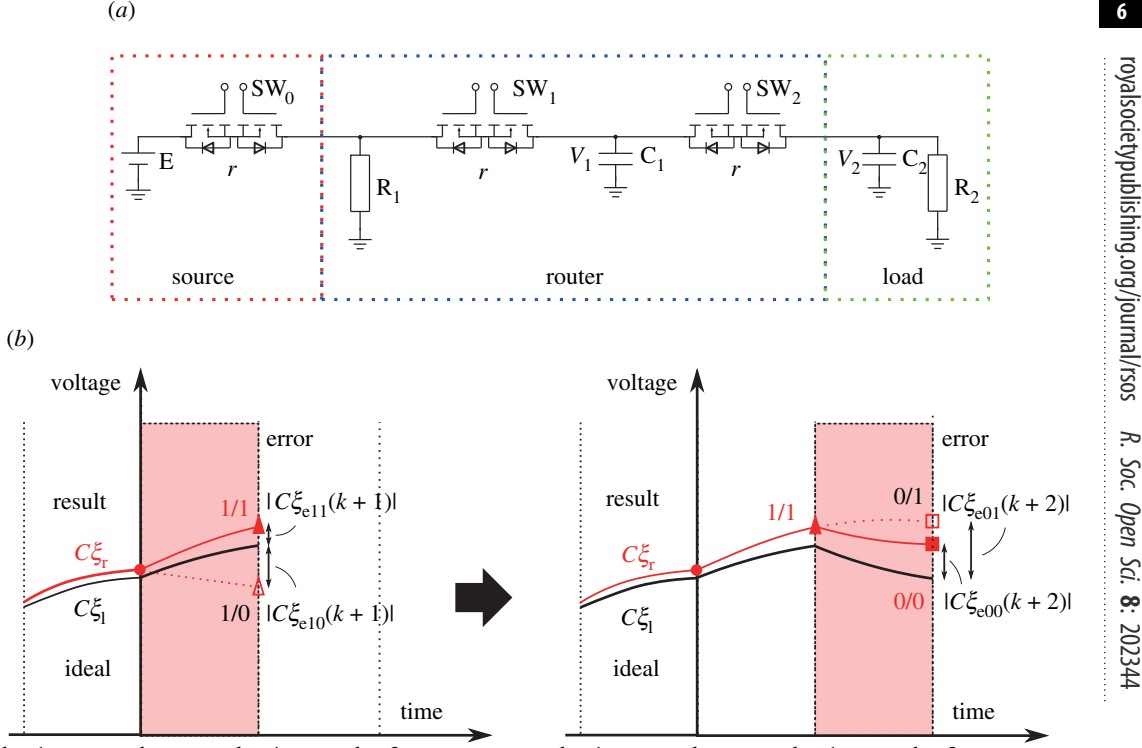

**Figure 4.** Verification of the error correction: (*a*) target circuit and (*b*) error correction algorithm.

The simulation is performed by using Simulink. Similar to the experiment, the power packet is sent to input ports with an information tag, which is assumed to include the request of the logic operation. Figure 3*c* demonstrates the result of simulation of the NAND operation; from the top to bottom, input voltage, output voltage, input power, output power and voltage of the capacitor. The logical change shows the NAND operation. The heat map shows the power transfer in the time slot. This means that all the input patterns '00', '01', '11' and '10' are operated to '1', '1', '0' and '1', respectively. They ensure that the experimental results follow the theoretical change of voltage, current and power with the logic operation, precisely.

The same as NAND operation, other possible logic operations are simulated and shown in figure 3*d*, which shows the heat map of power after the logic operation. It implies that all the logic operations of power packets can process the power in the defined time slots.

# 4. Error correction: dynamic selection of logic operation

In the power packet distribution network, the power packets produced by power sources exhibit a variety of voltages. It directly depends on the output of power sources or the charge of capacitances in the routers. The logic signal is expressed by a binary symbol based on the magnitude compared to the threshold. However, the electric power cannot be expressed as a symbol but is an accumulated physical quantity. The single power packet is defined at the duration of time required for power accumulation.

Figure 4*a* shows the schematic circuit of the power packet distribution network. The switch $SW_0$, attached to the source, corresponds to the mixer.

The power source assumed to have a random time-varying profile. It receives requests for power according to the load profile. To adjust power supply and demand, there is a possibility to select an appropriate logic operation at a router. The selection works simultaneously with the error correction of power in the logic operation of the packets. The error of power may appear during the time interval of the operation because of the behaviour of power sources, parasitic circuit elements, changes in loads, circuit configuration, misfirings of switches and so on. Some of the above-mentioned causes are predictable, where others are not.

The necessity of employing the error correction algorithm becomes obvious when there is difference between the power demand and the discretized power supplied in the packets.

Now, consider the case where the router can change the given logic operation of input packets to fulfil the load demand. The main focus is on the unary operation because the binary operation obeys the same algorithm.

When the router input '0' meets the demand '1', the router dispatches a power packet from the capacitor and the output becomes '1'. Although the logic operation logically satisfies the request '1', there may occur a difference between the output powers compared with the ideal case of the input. As mentioned, the difference causes error in the load voltage. Therefore, we design an algorithm to select a logic operation by taking this error into account.

To satisfy the load demand on each time slot, the unary logic operation proceeds. The router selects 'NOT' or 'Through' so that the load voltage reaches to the closest to its requirement. The router can exchange a part of the logic operation as shown in figure 4b.

When the selection of operations is expressed by the rule, the error correction achieves the appropriate power packet distribution without any feedback control of power flow. When the external disturbance such as current leak or voltage sag occurs in the network, the error correction works as well. Even when the precise train of power packets are sent from power sources, these disturbances may change the input of the power packet at the router, which manages the power supplied to the load. This is analogous to the error correction of signal based on the checksum.

Algorithm of the error correction cannot be defined uniquely. It is assumed that the router knows the circuit parameters of both the adjacent packet sources and loads. Then the router is governed by the internal model shown in figure 4a. The model is given as follows:

$$P_{11}: \begin{cases} V_1'(t) = \frac{1}{rC_1}\left(\frac{1}{(r/R_1)+2} - 2\right)V_1(t) + \frac{1}{rC_1}V_2(t) + \frac{1}{rC_1((r/R_1)+2)}E \\ V_2'(t) = \frac{1}{rC_2}V_1(t) - \frac{1}{rC_2}\left(1 + \frac{r}{R_2}\right)V_2(t) \end{cases}, \tag{4.1}$$

$$P_{00}: \begin{cases} V_1'(t) = 0 \\ V_2'(t) = -\frac{1}{R_2C_2}V_2(t) \end{cases}' \tag{4.2}$$

$$P_{10}: \begin{cases} V_1'(t) = -\frac{1+(r/R_1)}{rC_1((r/R_1)+2)}V_1(t) + \frac{1}{rC_1((r/R_1)+2)}E \\ V_2'(t) = -\frac{1}{R_2C_2}V_2(t) \end{cases} \tag{4.3}$$

and

$$P_{01}: \begin{cases} V_1'(t) = -\frac{1}{rC_1}V_1(t) + \frac{1}{rC_1}V_2(t) \\ V_2'(t) = \frac{1}{rC_2}V_1(t) - \frac{1}{rC_2}\left(1 + \frac{r}{R_2}\right)V_2(t) \end{cases}' \tag{4.4}$$

where $P_{ij}(i, j \in \{1, 0\})$ is equation corresponding to the case in which the logic of the router's input and output are $i$ and $j$, respectively. That is, $P_{00}$ and $P_{11}$ correspond to 'Through', and $P_{01}$ and $P_{10}$ to 'NOT'.

The algorithm is also applied dynamically in the packet distribution network. With time slot $T$ set as the unit time of discrete time $k$, $P_{ij}$ is discretized to be $\xi(k+1) = A_{ij}\xi(k) + B_{ij}E$, where $\xi(t) = [V_1(t)\ V_2(t)]^T$ is set as the voltage of each capacitor. With the constant matrix $C = [0\ 1]$, the load voltage is given by $C\xi(k)$. The error between the demand and the state is denoted by $\xi_e(k) = \xi_r(k) - \xi_l(k)$, where $\xi_l(k)$ represents target state in which the output of the source coincides with the load requirements, and $\xi_r(k)$ is actual state caused by random source outputs.

Following the model of the unary operation, the router selects a logic operation in a feed-forward manner. There are two possible logic operations as 'NOT' and 'Through' for input at each time slot. When an input is '1', the system follows either $P_{10}$ or $P_{11}$, and when it is '0', the system follows either $P_{01}$ or $P_{00}$. Therefore, when input is '1', the error is estimated as below

$$Q_{d\ i=1}: \begin{cases} \xi_l(k+1) = A_{11}\xi_l(k) + B_{11}E \\ \xi_r(k+1) = A_{1j}\xi_r(k) + B_{1j}E \\ \xi_{e1j}(k+1) = \xi_r(k+1) - \xi_l(k+1) \end{cases}. \tag{4.5}$$

Moreover, when it is '0', the error is estimated as below

$$Q_{d\ i=0}: \begin{cases} \xi_l(k+1) = A_{00}\xi_l(k) + B_{00}E \\ \xi_r(k+1) = A_{0j}\xi_r(k) + B_{0j}E \\ \xi_{e0j}(k+1) = \xi_r(k+1) - \xi_l(k+1) \end{cases}. \tag{4.6}$$

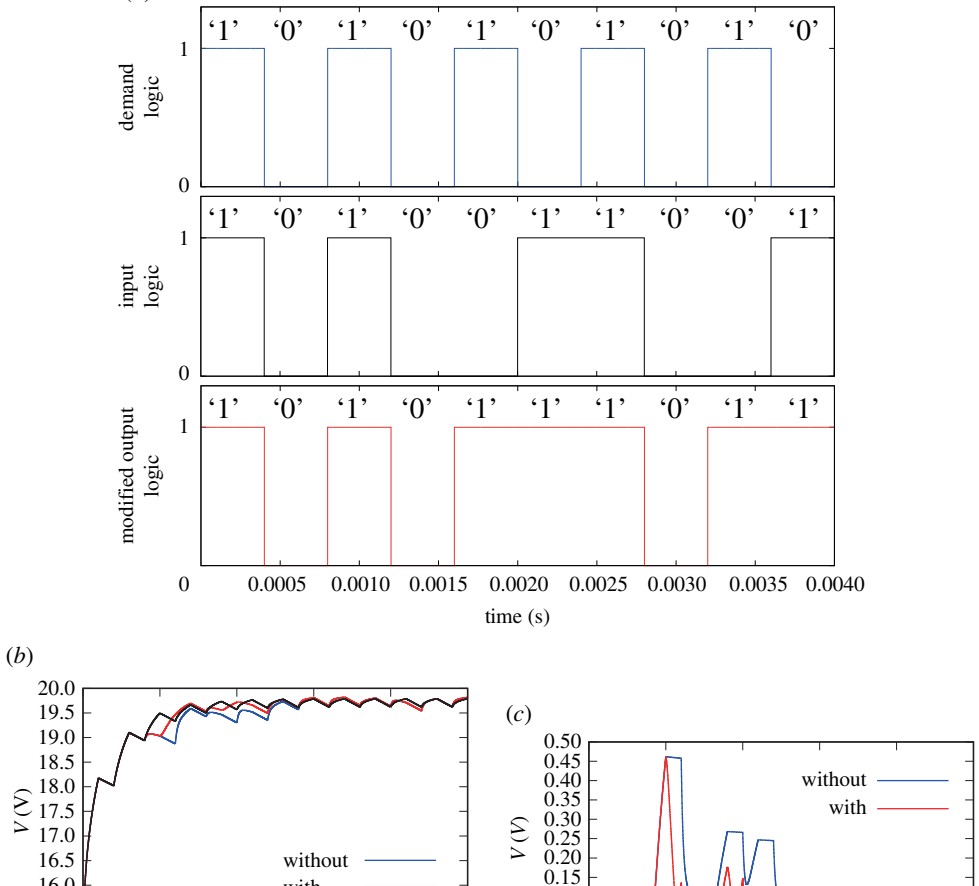

**Figure 5.** Simulation of error correction: (a) random input of the power packet and output using the proposed algorithm, (b) load voltage transition with and without the algorithm and (c) error of the load voltage.

As shown in figure 5a, the router selects the logic operation that produces the smallest output error at the next time slot. The selection is made under the following conditions:

$$s_{i=1}: \begin{cases} \text{NOT} & \text{if } |C\xi_{e10}(k+1)| < |C\xi_{e11}(k+1)| \\ \text{Through} & \text{otherwise} \end{cases} \tag{4.7}$$

and

$$s_{i=0}: \begin{cases} \text{NOT} & \text{if } |C\xi_{e01}(k+1)| < |C\xi_{e00}(k+1)| \\ \text{Through} & \text{otherwise} \end{cases}. \tag{4.8}$$

This algorithm selects a logic operation to achieve the least cumulative error.

The time slot is set at $T = 400\,\mu s$, the source voltage at 20 V, the initial voltage of the router capacitor at 18 V and the initial voltage of the load capacitor at 15 V. The circuit elements are set as described in table 2. The load is able to require a power packet at any arbitrary time. It requests the synchronization of the system clock. For simplicity, the timing of the requirement is set at every two time slots, which leads to keeping away from discussing the mechanism. In brief, the load requires '1' or '0', repeatedly. The input is provided at a random sequence.

The trial with algorithm (shown as 'with') is compared with the non-operated result (shown as 'without'). The latter implies that the router follows the same logic as the load requirement. In this study, $\overline{|C\xi_{e\text{without}}|}$ and $\overline{|C\xi_{e\text{with}}|}$ are set as the averaged voltage errors between the load requirement (target) and the output of 'without' and 'with' algorithm, respectively. Even in the case of random input, the router operates to erase the error of power. The capability depends on the probability of the

**Table 2.** Parameter of each element.

| MOSFET | $r = 22\,\text{m}\Omega$ |
| --- | --- |
| load for measurement | $R_1 = 1.5\,\text{k}\Omega$ |
| capacitor in the router | $C_1 = 4700\,\mu\text{F}$ |
| capacitor in load | $C_2 = 4700\,\mu\text{F}$ |
| resistive load | $R_2 = 10\,\Omega$ |

packet appearance. The output becomes close to the target value and fills the sum of temporal change of the electric power.

It is obvious that the binary operation works similarly for the error correction by a combination of logic operations. There are many possibilities for the combinations of operations. The error correction algorithm was verified in sending as many power packets as possible to satisfy the load demand in the power distribution network.

# 5. Conclusion

The electric power has been treated to be continually flowing according to the Kirchhoff's Law. Once the power is added in the circuit, the power cannot be operated and processed in the transmission line. The power packet distribution opened the possibility of processing a power unit by using the logic operation. The electric power can logically be operated in the form of power packet, which is physically a generated pulsed power attached to an information tag.

The power packetization seems to be considered as electric power digitization, but the power cannot be managed except from the density of the packet in the time duration. This is a continuous method by an averaged model. The reason is that the power is not digitally processed in the logic operation. The proposed method realizes the logic operation of the power packet. The proposed logic operation also makes it possible to process electric power as the digital unit. The simulation and experimental investigations prove that the unary and binary logic operations are physically possible by the switching circuit with storage.

The last part of the discussion was concentrated on the error correction in the logic operation of the power packet. The disturbances to the system cause the error between supplying power packets and the load demand. The disturbances include variety of power sources, wrong logic operation in the path, and appearance of the power loss on the transmission route of the power packet. This research work proposed the error correction algorithm to fill the demand similar to the checksum method by the logic operation.

The logic operation of the power packet and the error correction algorithm in the network achieve the fundamental basis of power processing.

Data accessibility. Datasets of experiments, simulation results and simulations codes are accessible from Kyoto University Research Information Repository at: https://doi.org/10.14989/260947.

Authors' contributions. T.H. designed the power packet network and proposed the concept of power processing using logic operation. S.M. designed the error correction algorithm. S.I. designed the circuits and the logic operations. All authors gave final approval for publication.

Competing interests. We declare we have no competing interests.

Funding. This research work was partially supported by JST-OPERA Program grant no. JPMJOP1841, Japan and the Grant-in-Aid for Scientific Research(B) 20H02151 from Japan Society for the Promotion of Science.

Acknowledgements. The authors thank the members of the Advanced Electric Systems Theory Laboratory, Kyoto University, for the fruitful discussion.

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
