## [Peer Review File · Royal Society Open Science]

Review History

RSOS-202344.R0 (Original submission)

Review form: Reviewer 1

Is the manuscript scientifically sound in its present form?

Yes

Are the interpretations and conclusions justified by the results?

Yes

Is the language acceptable?

Yes

Do you have any ethical concerns with this paper?

No

Have you any concerns about statistical analyses in this paper?

No

Recommendation?

Accept with minor revision (please list in comments)

Comments to the Author(s)

The authors have addressed the comments I raised for the previous submission of this work. Most answers are acceptable and clarified my doubts. In particular, I appreciate now that the proposed technique is aimed at stand-alone small low-voltage systems (such as EVs or satellites). This clarifies most issues but also creates some new doubts. If the system is actually isolated and relatively small, how redundant could possibly be the "power packet network"? I expect that in a EV there would not be really too many routers to allow many alternative paths that the the power can follow to reach the load. The following question is about the advantages of the proposed approach shows compared to conventional continuous power flows. If I understand well from the authors answers and assuming that the proposed approach works the better the more the redundancy of routers and connections, then the proposed approach is certainly more expensive than a conventional one. My question thus is: are the technical benefits good enough to make this approach interesting for practical applications not just laboratory experiments?

Review form: Reviewer 2

Is the manuscript scientifically sound in its present form?

Yes

Are the interpretations and conclusions justified by the results?

Yes

Is the language acceptable?

Yes

Do you have any ethical concerns with this paper?

No

Have you any concerns about statistical analyses in this paper?

No

Recommendation?

Accept with minor revision (please list in comments)

Comments to the Author(s)

Since this is a resubmission of a previously rejected paper, I have focused on the previous comments and how the authors have addressed them. In my view, the idea is fundamentally new for power processing, and the resubmission actually has addressed the previous comments adequately. As far as I know, the proposed power processing method is originated from the group of author, and this work addresses the implementational aspects which can be considered as realization steps. For this, I am happy to see it publish, as this may add to the literature providing a more complete picture of the proposed power processing concept. Afterall, the previous comments were legitimate and have been adequately addressed in this resubmission.

The paper however would benefit from some language editing for text improvement. For instance, in the Introduction, it is not clear what "at the edge" is referring to. Also, "Apart from the common sense" does sound odd if it only means "furthermore"! Though the paper is generally very readable, the authors are advised to go through a proper copyediting to improve the text.

Decision letter (RSOS-202344.R0)

Dear Professor Hikihara,

On behalf of the Editors, we are pleased to inform you that your Manuscript RSOS-202344 "Electric Power Processing Using Logic Operation and Error Correction" has been accepted for publication in Royal Society Open Science subject to minor revision in accordance with the referees' reports. Please find the referees' comments along with any feedback from the Editors below my signature.

Please submit your revised manuscript and required files (see below) no later than 7 days from today's (ie 25-May-2021) date. Note: the ScholarOne system will 'lock' if submission of the revision is attempted 7 or more days after the deadline. If you do not think you will be able to meet this deadline please contact the editorial office immediately.

on behalf of Professor R. Kerry Rowe (Subject Editor)
openscience@royalsociety.org

Associate Editor Comments to Author:

Thank you for your submitting your manuscript to Royal Society Open Science, which was previously reviewed at Proceedings A. As you can see, the reviewers are largely happy with the revisions made, and request some minor edits. We will be happy to accept your manuscript for publication once these have been addressed. Please ensure to provide a point-by-point response to the remaining comments when submitting your revised paper.

We also note one remaining comment from Reviewer 1, in that the data and code you have provided is not particularly clear: (<https://repository.kulib.kyoto-u.ac.jp/dspace/handle/2433/26094>). Please address this, and ensure all necessary data and code

are available in a readable format. Raw data and any additional code should be made available for an interested reader to attempt a replication of your study. This can be provided as supplementary material, or deposited to Dryad (which the journal is integrated with).

Finally, as you have been requested to edit the written English, you must provide proof that you have done so: acceptable proof includes a certificate of language-editing from a language editing service or a signed letter from a native speaker of English. We recommend you have a look at the language editing services endorsed by the Royal Society:

<https://royalsociety.org/journals/authors/language-polishing/>

Reviewer comments to Author:

Reviewer: 1

Comments to the Author(s)

The authors have addressed the comments I raised for the previous submission of this work. Most answers are acceptable and clarified my doubts. In particular, I appreciate now that the proposed technique is aimed at stand-alone small low-voltage systems (such as EVs or satellites). This clarifies most issues but also creates some new doubts. If the system is actually isolated and relatively small, how redundant could possibly be the "power packet network"? I expect that in a EV there would not be really too many routers to allow many alternative paths that the the power can follow to reach the load. The following question is about the advantages of the proposed approach shows compared to conventional continuous power flows. If I understand well from the authors answers and assuming that the proposed approach works the better the more the redundancy of routers and connections, then the proposed approach is certainly more expensive than a conventional one. My question thus is: are the technical benefits good enough to make this approach interesting for practical applications not just laboratory experiments?

Reviewer: 2

Comments to the Author(s)

Since this is a resubmission of a previously rejected paper, I have focused on the previous comments and how the authors have addressed them. In my view, the idea is fundamentally new for power processing, and the resubmission actually has addressed the previous comments adequately. As far as I know, the proposed power processing method is originated from the group of author, and this work addresses the implementational aspects which can be considered as realization steps. For this, I am happy to see it publish, as this may add to the literature providing a more complete picture of the proposed power processing concept. Afterall, the previous comments were legitimate and have been adequately addressed in this resubmission.

The paper however would benefit from some language editing for text improvement. For instance, in the Introduction, it is not clear what "at the edge" is referring to. Also, "Apart from the common sense" does sound odd if it only means "furthermore"! Though the paper is generally very readable, the authors are advised to go through a proper copyediting to improve the text.

===PREPARING YOUR MANUSCRIPT===

===PREPARING YOUR REVISION IN SCHOLARONE===

- If you are requesting a discretionary waiver for the article processing charge, the waiver form must be included at this step.
- If you are providing image files for potential cover images, please upload these at this step, and inform the editorial office you have done so. You must hold the copyright to any image provided.
- A copy of your point-by-point response to referees and Editors. This will expedite the preparation of your proof.

- Ensure that your data access statement meets the requirements at <https://royalsociety.org/journals/authors/author-guidelines/#data>. You should ensure that you cite the dataset in your reference list. If you have deposited data etc in the Dryad repository, please only include the 'For publication' link at this stage. You should remove the 'For review' link.
- If you are requesting an article processing charge waiver, you must select the relevant waiver option (if requesting a discretionary waiver, the form should have been uploaded at Step 3 'File upload' above).
- If you have uploaded ESM files, please ensure you follow the guidance at <https://royalsociety.org/journals/authors/author-guidelines/#supplementary-material> to include a suitable title and informative caption. An example of appropriate titling and captioning may be found at https://figshare.com/articles/Table_S2_from_Is_there_a_trade-off_between_peak_performance_and_performance_breadth_across_temperatures_for_aerobic_scope_in_teleost_fishes_/3843624.

Author's Response to Decision Letter for (RSOS-202344.R0)

See Appendix A.

Decision letter (RSOS-202344.R1)

Dear Professor Hikihara,

I am pleased to inform you that your manuscript entitled "Electric Power Processing Using Logic Operation and Error Correction" is now accepted for publication in Royal Society Open Science.

You can expect to receive a proof of your article in the near future. Please contact the editorial office (openscience@royalsociety.org) and the production office (openscience_proofs@royalsociety.org) to let us know if you are likely to be away from e-mail contact – if you are going to be away, please nominate a co-author (if available) to manage the proofing process, and ensure they are copied into your email to the journal. Due to rapid publication and an extremely tight schedule, if comments are not received, your paper may experience a delay in publication.

on behalf of Prof R. Kerry Rowe (Subject Editor)
openscience@royalsociety.org

Appendix A

Reply to Editors and Referees

— Comments by Editor —

Thank you for your submitting your manuscript to Royal Society Open Science, which was previously reviewed at Proceedings A. As you can see, the reviewers are largely happy with the revisions made, and request some minor edits. We will be happy to accept your manuscript for publication once these have been addressed. Please ensure to provide a point-by-point response to the remaining comments when submitting your revised paper.

We also note one remaining comment from Reviewer 1, in that the data and code you have provided is not particularly clear: (<https://repository.kulib.kyoto-u.ac.jp/dspace/handle/2433/26094>).

Please address this, and ensure all necessary data and code are available in a readable format. Raw data and any additional code should be made available for an interested reader to attempt a replication of your study. This can be provided as supplementary material, or deposited to Dryad (which the journal is integrated with).

Finally, as you have been requested to edit the written English, you must provide proof that you have done so: acceptable proof includes a certificate of language-editing from a language editing service or a signed letter from a native speaker of English. We recommend you have a look at the language editing services endorsed by the Royal Society: <https://royalsociety.org/journals/authors/language-polishing/>

We appreciate your support to review our submitted paper RSOS-202344. In the following, we will respond to the comments by reviewers one by one.

We are sorry for making the mistake to show the address of data and code. The correct address was :

<https://doi.org/10.14989/260947>

The previous address was lack of last "7".

In addition, we have asked ENAGO, which was listed on the home page of the Royal Society for language editing, at the first submission. However, we asked for the revised manuscript again. We are attaching the certification by ENAGO.

Once again, thank you for your valuable comments and suggestions.

The authors have addressed the comments I raised for the previous submission of this work. Most answers are acceptable and clarified my doubts. In particular, I appreciate now that the proposed technique is aimed at stand-alone small low-voltage systems (such as EVs or satellites). This clarifies most issues but also creates some new doubts. If the system is actually isolated and relatively small, how redundant could possibly be the "power packet network"? I expect that in a EV there would not be really too many routers to allow many alternative paths that the the power can follow to reach the load. The following question is about the advantages of the proposed approach shows compared to conventional continuous power flows. If I understand well from the authors answers and assuming that the proposed approach works the better the more the redundancy of routers and connections, then the proposed approach is certainly more expensive than a conventional one. My question thus is: are the technical benefits good enough to make this approach interesting for practical applications not just laboratory experiments?

We appreciate your effort to review our paper. We were happy to receive your new questions to confirm our idea. The questions are the substantial points of the power packet dispatching system. Of course, an EV is an example of an isolated system. We agree that we need many routers to manage all-electric power in EVs by power packets. The problems of EVs are the weight of the wire harness. They are divided into high power harness and low power harness. Each load needs independent wire connections. This is one of the disadvantages of the conventional electric system of the vehicle. However, the power packet dispatching system will give an idea of an active wire harness to manage the power distribution apart from the static wire connection. We do not think the power packet dispatching system should replace the whole power system in EVs. However, the method may give reductions of the weight of wires, which can reduce the capacity of the battery or achieve the extension of the operation.

We would like to show one example. It has been already known that a similar concept of active circuit wires was accidentally used in the space explorer named "Hayabusa," which achieved the complete return to earth from the asteroid, Itokawa. They managed the electric power in the explorer to operate the possible path of electricity in the last moment of return, because

of the lack of power sources and the trouble of circuit elements. In such a case we have the change of paths also worked.

In reference [13], it was confirmed that the regenerative power of motors can be managed by the router with storage. In reference [14] the convergence of several packets is regulated by the control of the router. In reference [15], the power packets can drive the motors with more than 1 kW. Based on the results, we can say that the system is enough ability to operate the auxiliary system, including door mirrors, LED lights, power windows, and so on. However, the power of the main driving and the safety system is over the present system capability. Therefore, we frankly answer that we cannot change the whole electric system of EV into the proposed system.

As for the price of the system construction, it seems a trade-off to the total cost of the system. Accumulation of converters at the edge must be of a similar capacity to routers. However, we can easily guess that the conventional converters are cheap enough. In addition, our proposed power packet system is under consideration by companies and projects in Japan. We are expecting development from the commercial side. Once again, we appreciate your valuable comments.

We add a sentence to answer your questions: "The following question is about the advantages of the proposed approach shows compared to conventional continuous power flows," and "are the technical benefits good enough to make this approach interesting for practical applications, not just laboratory experiments?" We thank you for this suggestion, because the revision strengthens this paper.

A page 2, line 27–32:

From the view point of applications, the power packet dispatching system enables us to reduce the wire connections between the power sources and loads depending on the change of demand and supply. It will be an advantage for the weight reduction of the harness in the closed systems. In addition, the system accepts the multipower sources in the single system to mitigate the increase of demands or to ensure safety without the new layout of the harness.

Since this is a resubmission of a previously rejected paper, I have focused on the previous comments and how the authors have addressed them. In my view, the idea is fundamentally new for power processing, and the resubmission actually has addressed the previous comments adequately. As far as I know, the proposed power processing method is originated from the group of author, and this work addresses the implementational aspects which can be considered as realization steps. For this, I am happy to see it publish, as this may add to the literature providing a more complete picture of the proposed power processing concept. After all, the previous comments were legitimate and have been adequately addressed in this resubmission.

The paper however would benefit from some language editing for text improvement. For instance, in the Introduction, it is not clear what "at the edge" is referring to. Also, "Apart from the common sense" does sound odd if it only means "furthermore"! Though the paper is generally very readable, the authors are advised to go through a proper copyediting to improve the text.

We appreciate your effort on our resubmitted paper. Your evaluation encourages us very much to proceed with our research.

According to the suggestions, we ask the language editing services of ENAGO, which is in the list of The Royal Society HP. They are shown by red marked parts.

Additional Revision

The referenced No. 5,

"J. Wang, et al., Spontaneous synchrony in power-grid networks, IEEE Industrial Electronics Magazine, 3(2), 16 (2009)."

was revised to

"J. Wang, et al., Smart Grid Technologies, IEEE Industrial Electronics Magazine, 3(2), 16 (2009)."

CERTIFICATE OF EDITING

This is to certify that the paper titled Electric Power Processing Using Logic Operation and Error Correction commissioned to us by Takashi Hikihara has been edited for English language, grammar, punctuation, and spelling by Enago, the editing brand of Crimson Interactive Pvt. Ltd under Normal Editing B2C.

- ✓ **ISO 17100:2015**
Translation Service Providers
- ✓ **ISO 27001:2013**
Information Security Management System
- ✓ **ISO 9001:2015**
Quality Management System

Issued by: Enago, Crimson Interactive Pvt. Ltd.
1001, Techniplex - II, S. V. Road,
Goregaon (W), Mumbai 400062, India.
Phone: 03-5050-5374
Fax: 03-4496-4934

Disclaimer : The intent of the author's message has been preserved during the editing process. The author is free to accept or reject our changes in the document after reviewing our edits. This certificate has been awarded at the time of sharing the final edited version (full file or sections of the file) with the author. Enago does not bear any responsibility for any alterations done by the author to the edited document post 4 Jun 2021.

Japan www.enago.jp, www.ulatus.jp, www.voxtab.jp
Taiwan www.enago.tw, www.ulatus.tw
China www.enago.cn, www.ulatus.cn
Brazil www.enago.com.br, www.ulatus.com.br
Germany www.enago.de

Russia www.enago.ru
Arabic www.enago.ae
Turkey www.enago.com.tr
S. Korea www.enago.co.kr
Global www.enago.com, www.ulatus.com, www.voxtab.com

About Crimson:
Crimson Interactive pvt ltd is one of the world's leading academic research support services. Since 2005, we've supported over 2 million researchers in 125 countries with their publication goals.

Subject Areas:

electric engineering, power and energy system

Keywords:

Electric power processing, Power packet, Logic operation, Error correction, Cyber-Physical system

Author for correspondence:

Takashi Hikihara

e-mail:

hikihara.takashi.2n@kyoto-u.ac.jp

Electric Power Processing Using Logic Operation and Error Correction

Shota Inagaki¹, Shiu Mochiyama¹ and Takashi Hikihara¹

¹Department of Electrical Engineering, Kyoto University, Katsura, Nishikyo, Kyoto 615-8510, Japan

In this study, electric power is processed using the logic operation method and the error correction algorithms to meet load demand. Electric power was treated as **the physical** flow through the distribution network, which was governed by circuit configuration and efficiency. The hardware required to digitize or packetize electric power, which is called power packet router, was developed in this research work for low power distribution. It provides **an** opportunity for functional electric power dispatching **while** disregarding the power flow in the circuit. This study proposes a new design for the network, which makes the logic operation of electric power possible and provides an algorithm to correct the inaccuracies caused by dissipation and noise. Phase shift of the power supply network is resulted by implementing the introduced design.

1. Introduction

Electric power is considered as a continually flowing quantity caused by potential difference between nodes in an electric circuit. The smart grid concept interconnects power engineering with information and communication technology (ICT). It also acknowledges the regulation of the power flow between power sources and loads. However, power keeps continually flowing according to an electric circuit configuration even when load demand is quickly and forcibly altered by ICT systems. This fact in turn might lead to a spillover of demands. Hence, in cyber-physical systems, simultaneous operation of the electric network and the ICT system is essential.

Conventional AC power transmission system is an interesting complex network [1,2]. The power network maintains the rule of just-in-time for the system stability and regulation purposes based on dynamics and efficiency. The time constant of the system dynamics is usually long enough to satisfy the temporal change of load demands at the edge. The supply and demand discrepancy is absorbed in huge capacity of linked generators to keep the equilibrium between the generators in the vicinity. Here, we will reconsider the demand and response system at low power distribution, especially in the closed and isolated systems.

This paper focuses the power distribution in a local grid of a closed system isolated from the commercial power grid, e.g., vehicles, robots, airplanes, and satellites. In such systems, it is difficult to adopt the aforementioned method for huge power systems. Closed systems have several power sources and batteries, including regeneration from loads that must be operated simultaneously. Thus, there is a limitation of power capacity and the request for real-time power control. In these systems, the information and power must be supplied to the edge without conflict.

In the 1990s, two significant studies were reported in the literature: one was a method to trace the electric power flow in the network [3] and the other was the power packet dispatching method [4]. A smart grid concept provides an opportunity to implement the former method [5], and recently the peer to peer technology along with the block chain is introduced to gain the incentive of employing renewable energy sources [6]. These methods intend to alleviate the problem of mismatch between physical power delivery and cyber analysis.

However, the concept of power packet dispatching was too early to be recognized in the 1990s, because energy storage systems, power-switching devices, and ICT were insufficiently nature. Since the late 2000s, the concept has been studied theoretically and experimentally [7–12]. Besides theoretical considerations, the development of bidirectional routers in kW order [13], the congestion management of the power packet dispatching [14], and the kW class motor drives [15] were achieved. These studies revealed that there is inadequate interface to link the cyber systems and operation of an electric power network. From the view point of applications, the power packet dispatching system enables us to reduce the wire connections between the power sources and loads depending on the change of demand and supply. It will be an advantage for the weight reduction of the harness in the closed systems. In addition, the system accepts the multipower sources in the single system to mitigate the increase of demands or to ensure safety without the new layout of the harness.

This study proposes and confirms *power processing* using logic operation and error correction algorithm, which is not the same as the information processing [16]. The reason lies in the fact that power is not a symbol but a real physical quantity. Detailed discussion as to the discrepancy between power processing and information processing has already been reported in [17]. It confirms that the power packet is logically processed by the unary and binary symbols corresponding to the existence of the power payload attached with an information tag. Furthermore, the payload of power must obey the law of conservation of energy and the law of dissipation. Here, it is crucial to maintain simultaneous of information and power flows.

This study follows the definition of the logic operation of the power packet dispatching and the algorithm of error correction. The proposed method creates a completely different power distribution system, which disregards the power flow governed by the circuit configuration.

2. The logic operation of electric power

Fig. 1(a) shows the structure of the power packet. It consists of an information tag and a payload of power. The information tag includes the header and the footer. They are attached as a voltage waveform without current. It means that the tag will not request the circuit configuration between source and load. The routers send and receive the information at sending the power packet. They transfer the payload as a pulsed power with load current, which is decided by circuit configuration. Like the concept, the header conveys a destination address for the payload of

Figure 1. Power packet and its dispatching system: (a) basic structure of the power packet, (b) schematic view of a router, and (c) power packet dispatching system.

power, the requested operation to the payload, and the control codes. The footer conveys an end control signal of the packet.

Fig. 1(b) shows the schematic view of a router. The router selects one of the inputs when the left switch turns ON. After the switch turns on and connects the source, the capacitor stores the power. The power is transmitted to the output by closing the right and opening left switch set in the desired direction. These switches are controlled according to the received information tag of the packet. Therefore, each power packet is initially tagged to the destination and the request of control. Then, the output ports are selected according to the information written on the header. The previous research works were carried out based on the setting. Fig. 1(c) shows the schematic of a power packet dispatching system, where the mixer generates a power packet and a router dispatches it according to the header.

The output side of the router exactly coincides with a mixer. The packetized power is dispatched by using the method of time division multiplex (TDM). The modulation method gives room to avoid the collision of the power packets. The rule of packetization was proposed in (9) based on the optimal dynamic quantizers for discrete-valued input control [17,18]. The rule has already been verified on motor drive in a robot arm with a feedback loop [19,20].

The logic operation is defined at each time slot concerning the existence (“1”) or non-existence (“0”) of the power packet according to the voltage amplitude of the payload compared to the threshold voltage. That is, “1” denotes the existence of the payload higher than the threshold voltage and “0” denotes the existence of the payload lower than the threshold. As we set the threshold voltage at $\epsilon \sim 0$ V higher than the noise, it shows the existence of a power packet.

The unary operation is applied to the input during a time slot T of the payload.

In this study, logic **operation** of power packets is defined as a function of the router. Logic operations are physically realized as unary (one-bit) or binary (two-bit) operation of power packets. The unary logic operations imply “NOT” and “Through.” The binary operations correspond to “AND,” “OR,” “NAND,” and so on. The logic of the signal defines the output of the operations as a logic signal “0” or “1.” However, *power processing* in this research is introduced as the logic operations of power depending on the existence of power. Then, the output becomes logically operated power. Let us assume that every power packet is handled synchronously at the same clock signal. The asynchronous packet transfer has already been discussed in [21]. In this study, command of operation is set on the header of the power packet.

The steps of a logic operation decided by the number of operands are shown here. As for the unary operation, **output is decided in the time slot, simultaneously. However**, for the binary operation, the output is postponed until a pair of two consecutive time slots are filled. Let us index them to f(oward) and b(ackward). Fig. 2(b) describes the binary operation. The output is discharged at the time slot b. This is the fundamental binary operation. There are plenty of variations of the operation depending on the number of ports and the time sequence of TDM.

The definition of the input/output logic relationship follows conventional operations. Fig. 2(a) shows the unary noninversion operation “Through” and inversion one “NOT.” Fig. 2(b) shows the binary operation “NAND” and “AND.” In the logic operation, each voltage is held until the end of an operation. However, the operations’ results do not hold the electric power carried by the input power packet because of the circuit configuration.

Electric power is given by temporal integration of instantaneous power by voltage times current. It appears only at the payload in a packet. The logic operation of power requires us to set an additional procedure using an *energy buffer*. An example of “NAND” operation is given in this study. Table 1 shows the input/output relationship and the usage of the buffer for “NAND.” The output is given at the backward time slot b. The input “1” at forwarding time slot f is always stored.

When the input is “1” at the time slot b and the calculated output becomes “0,” the input power is stored. In addition, when the input is “0” at the time slot b, and the output becomes “1,” the buffer is discharged. It satisfies that the power consistently flows according to the conventional logic rules. After the logic operation, the structure of the new power packet is configured as in Fig. 2(c), and follows TDM manners.

Table 1. Truth table of NAND.

input		output		buffer	
f	b	f	b	f	b
0	0	-	1	-	discharge
0	1	-	1	-	-
1	1	-	0	store	store
1	0	-	1	store	discharge

3. Verification of static logic operation by router

The logic operation of power is verified by conducting experiments and simulation of the circuit, as shown in Fig. 3(a). The mixer generates power packets and dispatches them to the router. The router gets the tag information and executes the logic operation designated by the header.

The load is set at $10\ \Omega$ and connected to the router output. The switching elements are composed of Si MOSFETs with $16\ \text{m}\Omega$ on-resistance, and SiC SBDs with $0.8\ \text{V}$ forward voltage drop. ZYBO Zynq-7000 development board is adapted as the controller of the switches. The power source is set at $18.4\ \text{V}$ DC and the storage is already charged.

Fig. 3(b) demonstrates the experimental results of the NAND operation. Depending on the inputs of the power packet, the logic operation is carried out. The experimental results are verified by simulation of the same circuit in Fig. 3(a). The current exactly flows according to the power packet. The null payload and the tags do not accept current flow. **However**, the power is packetized and NAND operation is achieved as the binary operation in Fig. 3(b). The logic operation of power packets achieves not only the signal operation but also the logic operation of power.

The simulation is performed by utilizing Simulink. Similar to the experiment, the power packet is sent to input ports with an information tag, which is assumed to include the request of the logic operation. Fig. 3(c) demonstrates the result of simulation of the NAND operation; from the top to

Figure 2. Allocation of logic value: (a) unary operation, (b) binary operations, and (c) packet for logic operation.

bottom, input voltage, output voltage, input power, output power, and voltage of the capacitor. The logical change shows the NAND operation. The heat map shows the power transfer in the time slot. This means that all the input patterns "00," "01," "11," and "10" are operated to "1," "1," "0," and "1" respectively. They ensure that the experimental results follow the theoretical change of voltage, current, and power with the logic operation, precisely.

The same as NAND operation, other possible logic operations are simulated and shown in Fig. 3(d), which shows the heat map of power after the logic operation. It implies that all the logic operations of power packets can process the power in the defined time slots.

4. Error correction: dynamic selection of logic operation

In the power packet distribution network, the power packets produced by power sources exhibit a variety of voltages. It directly depends on the output of power sources or the charge of capacitances in the routers. The logic signal is expressed by a binary symbol based on the magnitude compared to the threshold. **However**, the electric power cannot be expressed as a symbol but is an accumulated physical quantity. The single power packet is defined at the duration of time required for power accumulation.

Fig. 4(a) shows the schematic circuit of the power packet distribution network. The switch SW_0 , attached to the source, corresponds to the mixer.

The power source assumed to have a random time-varying profile. It receives requests for power according to the load profile. To adjust power supply and demand, there is a possibility to select an appropriate logic operation at a router. The selection works simultaneously with the error correction of power in the logic operation of the packets. The error of power may appear

Figure 3. Logic operation of the electric power: (a) experimental circuit for logic operation, (b) NAND operation by experimental circuit in (a), and (c) simulated switching pattern of NAND operation (input power, output power, and voltage of capacitor in the circuit in (a)). The color bar depicts the energy level at each time interval. The energy is normalized by the maximum value of the input energy, and (d) output power of possible logical operations of the power packets.

during the time interval of the operation because of the behavior of power sources, parasitic circuit elements, changes in loads, circuit configuration, misfirings of switches, and so on. Some of the above mentioned causes are predictable, where others are not.

The necessity of employing the error correction algorithm becomes obvious when there is difference between the power demand and the discretized power supplied in the packets.

Now, consider the case, where the router can change the given logic operation of input packets to fulfill the load demand. The main focus is on the unary operation because the binary operation obeys the same algorithm.

When the router input “0” meets the demand “1,” the router dispatches a power packet from the capacitor and the output becomes “1.” Although the logic operation logically satisfies the request “1,” there may occur a difference between the output powers compared with the ideal

case of the input. As mentioned, the difference causes error in the load voltage. Therefore, we design an algorithm to select a logic operation by taking this error into account.

To satisfy the load demand on each time slot, the unary logic operation is proceeded. The router selects “NOT” or “Through” so that the load voltage reaches to the closest to its requirement. The router can exchange a part of the logic operation as shown in Fig. 4(b).

When the selection of operations is expressed by the rule, the error correction achieves the appropriate power packet distribution without any feedback control of power flow. When the external disturbance such as current leak or voltage sag occurs in the network, the error correction works as well. Even when the precise train of power packets are sent from power sources, these disturbances may change the input of the power packet at the router, which manages the power supplied to the load. This is analogous to the error correction of signal based on the checksum.

Figure 4. Verification of the error correction: (a) target circuit, (b) error correction algorithm.

Algorithm of the error correction cannot be defined uniquely. It is assumed that the router knows the circuit parameters of both the adjacent packet sources and loads. Then the router is governed by the internal model shown in Fig. 4(a). The model is given as follows:

$$P_{11}: \begin{cases} V_1'(t) = \frac{1}{rC_1} \left(\frac{1}{\frac{r}{R_1} + 2} - 2 \right) V_1(t) + \frac{1}{rC_1} V_2(t) \\ + \frac{1}{rC_1} \left(\frac{1}{\frac{r}{R_1} + 2} \right) E \\ V_2'(t) = \frac{1}{rC_2} V_1(t) - \frac{1}{rC_2} \left(1 + \frac{r}{R_2} \right) V_2(t) \end{cases}, \quad (4.1)$$

$$P_{00}: \begin{cases} V_1'(t) = 0 \\ V_2'(t) = -\frac{1}{R_2 C_2} V_2(t) \end{cases}, \quad (4.2)$$

$$P_{10}: \begin{cases} V_1'(t) = -\frac{1+\frac{r}{R_1}}{rC_1(\frac{r}{R_1}+2)}V_1(t) + \frac{1}{rC_1(\frac{r}{R_1}+2)}E \\ V_2'(t) = -\frac{1}{R_2C_2}V_2(t) \end{cases}, \quad (4.3)$$

$$P_{01}: \begin{cases} V_1'(t) = -\frac{1}{rC_1}V_1(t) + \frac{1}{rC_1}V_2(t) \\ V_2'(t) = \frac{1}{rC_2}V_1(t) - \frac{1}{rC_2}\left(1 + \frac{r}{R_2}\right)V_2(t) \end{cases}. \quad (4.4)$$

Where $P_{ij}(i, j \in \{1, 0\})$ is equation corresponding to the case in which the logic of the router's input and output are i and j , respectively. That is, P_{00} and P_{11} correspond to "Through," and P_{01} and P_{10} to "NOT."

The algorithm is also applied dynamically in the packet distribution network. With time slot T set as the unit time of discrete time k , P_{ij} is discretized to be $\xi(k+1) = A_{ij}\xi(k) + B_{ij}E$. Where $\xi(t) = [V_1(t) V_2(t)]^T$ is set as the voltage of each capacitor. With the constant matrix $C = [0 \ 1]$, the load voltage is given by $C\xi(k)$. The error between the demand and the state is denoted by $\xi_e(k) = \xi_r(k) - \xi_l(k)$, where $\xi_l(k)$ represents target state in which the output of the source coincides with the load requirements, and $\xi_r(k)$ is actual state caused by random source outputs.

Following the model of the unary operation, the router selects a logic operation in a feed-forward manner. There are two possible logic operations as "NOT" and "Through" for input at each time slot. When an input is "1," the system follows either P_{10} or P_{11} , and when it is "0," the system follows either P_{01} or P_{00} . Therefore, when input is "1," the error is estimated as below:

$$Q_{d \ i=1}: \begin{cases} \xi_l(k+1) = A_{11}\xi_l(k) + B_{11}E \\ \xi_r(k+1) = A_{1j}\xi_r(k) + B_{1j}E \\ \xi_{e1j}(k+1) = \xi_r(k+1) - \xi_l(k+1) \end{cases}. \quad (4.5)$$

Moreover, when it is "0," the error is estimated as below:

$$Q_{d \ i=0}: \begin{cases} \xi_l(k+1) = A_{00}\xi_l(k) + B_{00}E \\ \xi_r(k+1) = A_{0j}\xi_r(k) + B_{0j}E \\ \xi_{e0j}(k+1) = \xi_r(k+1) - \xi_l(k+1) \end{cases}. \quad (4.6)$$

As shown in Fig. 5(a), the router selects the logic operation that produces the smallest output error at the next time slot. The selection is made under the following conditions:

$$s_{i=1}: \begin{cases} \text{NOT} & \text{if } |C\xi_{e10}(k+1)| < |C\xi_{e11}(k+1)| \\ \text{Through} & \text{otherwise} \end{cases}, \quad (4.7)$$

$$s_{i=0}: \begin{cases} \text{NOT} & \text{if } |C\xi_{e01}(k+1)| < |C\xi_{e00}(k+1)| \\ \text{Through} & \text{otherwise} \end{cases}. \quad (4.8)$$

This algorithm selects a logic operation to achieve the least cumulative error.

The time slot is set at $T = 400 \mu s$, the source voltage at 20V, the initial voltage of the router capacitor at 18 V, and the initial voltage of the load capacitor at 15 V. The circuit elements are set as described in Table 2. The load is able to require a power packet at any arbitrary time. It requests the synchronization of the system clock. For simplicity, the timing of the requirement is set at every two time slots, which leads to keep away from discussing the mechanism. In brief the load requires "1" or "0," repeatedly. The input is provided at a random sequence.

Table 2. Parameter of each element.

MOSFET	$r = 22m\Omega$
Load for measurement	$R_1 = 1.5 \text{ k}\Omega$
Capacitor in the Router	$C_1 = 4700 \mu\text{F}$
Capacitor in Load	$C_2 = 4700 \mu\text{F}$
Resistive load	$R_2 = 10 \Omega$

Figure 5. Simulation of error correction: (a) random input of the power packet and output using the proposed algorithm, (b) load voltage transition with and without the algorithm, and (c) error of the load voltage.

The trial with algorithm (shown as “with”) is compared with the nonoperated result (shown as “without”). The latter implies that the router follows the same logic as the load requirement. In this study, $|\overline{C\xi_{e\ without}}|$ and $|\overline{C\xi_{e\ with}}|$ are set as the averaged voltage errors between the load requirement (target) and the output of “without” and “with” algorithm, respectively. Even in the case of random input, the router operates to erase the error of power. The capability depends on the probability of the packet appearance. The output becomes close to the target value and fills the sum of temporal change of the electric power.

It is obvious that the binary operation works similarly for the error correction by a combination of logic operations. There are many possibilities for the combinations of operations. The error correction algorithm was verified in sending as much power packets as possible to satisfy the load demand in the power distribution network.

5. Conclusion

The electric power has been treated to be continually flowing according to the Kirchhoff's law. Once the power is added in the circuit, the power cannot be operated and processed in the transmission line. The power packet distribution opened the possibility of processing a power unit by using the logic operation. The electric power can logically be operated in the form of power packet, which is physically a generated pulsed power attached to an information tag.

The power packetization seems to be considered as electric power digitization, but the power cannot be managed except from the density of the packet in the time duration. This is a continuous method by an averaged model. The reason is that the power is not digitally processed in the logic operation. The proposed method realizes the logic operation of the power packet. The proposed logic operation also makes it possible to process electric power as the digital unit. The simulation and experimental investigations prove that the unary and binary logic operations are physically possible by the switching circuit with storage.

The last part of the discussion was concentrated on the error correction in the logic operation of the power packet. The disturbances to the system cause the error between supplying power packets and the load demand. The disturbances include variety of power sources, wrong logic operation in the path, and appearance of the power loss on the transmission route of the power packet. This research work proposed the error correction algorithm to fill the demand similar to the checksum method by the logic operation.

The logic operation of the power packet and the error correction algorithm in the network achieve the fundamental basis of power processing.

Ethics. Insert ethics text here.

Data Accessibility. [Datasets of experiments, simulation results, and simulations codes are accessible from https://doi.org/10.14989/260947](https://doi.org/10.14989/260947)

Authors' Contributions. T.H. designed the power packet network and proposed the concept of power processing using logic operation. S.M. designed the error correction algorithm. S.I. designed the circuits and the logic operations. All authors gave final approval for publication.

Competing Interests. We declare we have no competing interests.

Funding. This research work was partially supported by JST-OPERA Program Grant Number JPMJOP1841, Japan and the Grant-in-Aid for Scientific Research(B) 20H02151 from Japan Society for the Promotion of Science.

Acknowledgements. The authors thank the members of the Advanced Electric Systems Theory Laboratory, Kyoto University, for the fruitful discussion.

Disclaimer. Insert disclaimer text here.

References

1. Y. Susuki, I. Mezić, and T. Hikiyara, Coherent Swing Instability of Power Grids, *Journal of Nonlinear Science*, 21(3), 403 (2011).
2. A. E. Motter, S. A. Myers, M. Anghel, and T. Nishikawa, Spontaneous Synchrony in Power-Grid networks, *Nature Physics*, 9(3), 191 (2013).
3. J. Bialek, Tracing the Flow of Electricity, *IEE Proceedings: Generation, Transmission and Distribution*, 143(4), 4 (1996).
4. J. Toyoda and H. Saitoh, Proposal of an Open-Electric-Energy-Network (OEEN) to Realize Cooperative Operations of IOU and IPP, *Proceedings of 1998 International Conference on Energy Management and Power Delivery (EMPD ' 98)*, 1, 218 (1998).
5. J. Wang, et al., Smart Grid Technologies, *IEEE Industrial Electronics Magazine*, 3(2), 16 (2009).
6. G. Gao, et al., Application of Blockchain Technology in peer-to-peer Transaction of Photovoltaic Power Generation, *2nd IEEE Advanced Information Management, Communicates, Electronic and Automation Control (IMCEC2018)*, 2289 (2018).
7. T. Takuno, M. Koyama, and T. Hikiyara, In-Home Power Distribution Systems by Circuit Switching and Power Packet Dispatching, *Proceedings of First IEEE International Conference on Smart Grid Communications*, 427 (2010).
8. R. Takahashi, K. Tashiro, and T. Hikiyara, Router for Power Packet Distribution Network: Design and Experimental Verification, *IEEE Transactions on Smart Grid*, 6 (2), 618 (2015).
9. R. Takahashi, S. Azuma, and T. Hikiyara, Power Regulation with Predictive Dynamic Quantizer in Power Packet Dispatching System, *IEEE Transactions on Industrial Electronics*, 63 (12), 7653 (2016).
10. M. M. He, et al. An Architecture for Local Energy Generation, Distribution, and Sharing, *IEEE Energy 2030 Conference*, 1 (2008).
11. E. Gelenbe, Energy Packet Networks: Adaptive Energy Management for the Cloud, *Proceedings of 2nd International Workshop on Cloud Computing Platforms*, 1 (2012).
12. H. Sugiyama, Pulsed Power Network Based on Decentralized Intelligence for Reliable and Lowloss Electrical Power Distribution, *Journal of Artificial Intelligence and Soft Computing Research*, 5 (2), 97 (2015).
13. N. Yoshida, R. Takahashi, and T. Hikiyara, Power Regeneration from DC Motor with Bidirectional Router in Power Packet Dispatching System, *IEEE Transactions on Circuits and Systems II: Express Briefs*, 1, doi: 10.1109/TCSII.2020.2968384 (2020).
14. S. Katayama and T. Hikiyara, Power Packet Router with Power and Signal Switches for a Single Power Packet, *IEEE Transactions on Circuits and Systems II: Express Briefs*, 67(12), 3242, doi: 10.1109/TCSII.2020.2983444 (2020)
15. S. Mochiyama, T. Okuda and T. Hikiyara, Power Packet Dispatching with Shared Power Line: Experimental Verification for Industrial Applications, *IEEE Journal of Emerging and Selected Topics in Industrial Electronics*, 2(2), 164, doi: 10.1109/JESTIE.2020.3043141 (2020).
16. C. E. Shannon, A Mathematical Theory of Communication, *The Bell System Technical Journal*, 27, 379, 623 (1948).
17. S. Nawata, A. Maki, and T. Hikiyara, Power Packet Transferability via Symbol Propagation Matrix, *Proceedings Mathematical, Physical, and Engineering Science*, 474(2213), 20170552 (2018).
18. S. Azuma and T. Sugie, Optimal Dynamic Quantizers for Discrete-Valued Input Control, *Automatica*, 44 (2), 396 (2008).
19. S. Azuma and T. Sugie, Synthesis of Optimal Dynamic Quantizers for Discrete-Valued Input Control, *IEEE Transactions on Automatic Control*, 53 (9), 2064 (2008).
20. S. Mochiyama and T. Hikiyara, Packet-Based Feedback Control of Electrical Drive and Its Application to Trajectory Tracking of Manipulator, *International Journal of Circuit Theory and Applications*, 47 (4), 612 (2019).
21. Y. Zhou, R. Takahashi, N. Fujii, and T. Hikiyara, Power Packet Dispatching with Second-Order Clock Synchronization, *International Journal of Circuit Theory and Applications* 44(3), 729 (2016).